# In Vitro Effect of Drilling Speed on the Primary Stability of Narrow Diameter Implants with Varying Thread Designs Placed in Different Qualities of Simulated Bone

**DOI:** 10.3390/ma12081350

**Published:** 2019-04-25

**Authors:** Georgios E. Romanos, Daniel J. Bastardi, Rachel Moore, Apoorv Kakar, Yaro Herin, Rafael A. Delgado-Ruiz

**Affiliations:** 1Department of Periodontology, Laboratory for Periodontal-, Implant-, Phototherapy (LA-PIP), School of Dental Medicine, Stony Brook University, Stony Brook, New York, NY 11794, USA; Daniel.Bastardi@stonybrook.edu (D.J.B.); Rachel.moore@stonybrook.edu (R.M.); Apoorv.Kakar@stonybrook.edu (A.K.); yaroslav.herin@stonybrook.edu (Y.H.); 2Department of Prosthodontics and Digital Technology, School of Dental Medicine, Stony Brook University, Stony Brook, New York, NY 11794, USA; rafael.delgado-ruiz@stonybrook.edu

**Keywords:** drilling speed, implant stability quotient, narrow diameter implants, primary stability

## Abstract

It is hypothesized that there is no statistically significant impact of drilling speed (DS) on the primary stability (PS) of narrow-diameter implants (NDIs) with varying thread designs placed in dense and soft simulated bone. The aim of this in vitro study was to evaluate the impact of DS on the PS of NDIs with varying thread designs placed in dense and soft simulated bone. Two hundred and forty osteotomies for placement of various implant macro-designs were divided into three groups (80 implants per group): Group A (NobelActive, 3.0/11.5 mm); Group B (Astra OsseoSpeed-EV, 3.0/11 mm); and Group C (Eztetic-Zimmer, 3.1/11.5 mm) implants. These implants were placed in artificial dense and soft simulated bone using DSs of 800 and 2000 revolutions per minute (RPM). Resonance frequency analysis (RFA) and implant stability quotient (ISQ) were assessed. Group comparisons were performed using the one-way analysis of variance with Tukey’s post hoc tests. Level of significance was set at P < 0.05. In groups A and B, there was no difference in the ISQ for NDIs inserted in dense bone at 800 and 2000 RPM. In Group C, ISQ was significantly higher for NDIs placed in dense bone at 800 PRM compared to 2000 RPM (P < 0.05). In Group A, ISQ values were significantly higher for NDIs inserted in soft bone at 2000 RPM as compared to those inserted at 800 RPM (P < 0.05). For NDIs, a lower drilling speed in dense artificial simulated bone and a higher drilling speed in soft artificial simulated bone is associated with high primary stability.

## 1. Introduction

A critical factor that influences the long-term success and survival of dental implants is the achievement of primary stability (PS) at the time of implant insertion [1,2,3]. The PS of implants is related to mechanical anchorage with the surrounding bone, whereas secondary or biological stability is associated with peri-implant bone remodeling and regeneration [4]. Factors that influence PS include bone quantity and quality, surgical technique, and implant macro-design [5,6].

The quality and quantity of cortical and/or cancellous bone is a critical parameter that governs the overall outcome of dental implant therapy. Poor bone quality and/or quantity is associated with impairment in the peri-implant healing process and bone resorption, which if left untreated may progress to peri-implant diseases—such as peri-implantitis and implant failure [4,7]. The densest bone is usually present in the anterior mandible, followed by premaxilla and the posterior mandible [8]. The least dense bone is usually present in the posterior maxilla, as well as the mandible. The most common bone types in the anterior and posterior maxilla are type II and type IV, respectively [9]. In an in vitro study by Almeida et al. [3], the influence of drilling speed (DS) used during implant site preparation on PS was assessed. In this study, tapered designed implants were inserted at drilling speeds of 800 and 1500 RPM in solid rigid polyurethane foam and cellular rigid polyurethane foam, which simulated dense and soft bone, respectively [3]. The results showed that bone quality—not the DS—influences PS [3]. However, osteotomies performed at a higher DS (1500 PRM) have been associated with thermal changes that may jeopardize the integrity of peri-implant bone tissue [10]. 

The use of narrow diameter implants (NDIs) for the oral rehabilitation of partially and completely edentulous individuals is increasing [11,12]. One-year follow-up results of a prospective multicenter clinical study showed a high survival rate, stable bone levels, and healthy soft tissue around NDIs [13]. This study [13] concluded that NDIs are a predictable and safe treatment protocol for patients with limited bone volume. Moreover, a recent systematic review and meta-analysis of clinical studies concluded that oral rehabilitation using NDIs is a predictable treatment strategy [14,15], providing a promising treatment option for patients with substantial oral health-related quality of life and a positive attitude towards implant treatment for at least five years [16]. 

NDIs placed to support single crowns in the posterior region did not differ from regular diameter implants in regard to marginal bone levels, implant survival, and success rates [17]. Similarly, single crowns supported by NDIs in the anterior region are a predictable treatment, and comparable to treatment supported by regular-diameter implants. However, more studies are needed to evaluate the long-term esthetic outcomes of single crowns supported by NDIs [18]. In addition, resilient long-term data and data on the possible risk of biological and technical complications with wide platform teeth on NDIs are missing so far [19]. It is hypothesized that there is no statistically significant impact of DS on the PS of NDIs with varying thread designs placed in dense and soft simulated bone. The aim of the present in vitro experiment was to evaluate the impact of DS on the PS of NDIs placed in dense and soft simulated bone.

## 2. Methods

### 2.1. Surgical Protocol

A total of 240 osteotomies using a surgical implant unit (Frios S/i^®^ unit, Friadent, Mannheim, Germany) were created in this study (Figure 1). The implants were distributed into three separate groups: Group A (NobelActive implants, Nobel Biocare, Yorba Linda, CA, USA); Group B (Astra OsseoSpeed-EV implants, Dentsply-Sirona, York, PA, USA); and Group C (Eztetic-Zimmer implants, Zimmer Biomet Dental, Palm Beach Gardens, FL, USA). These implant designs were selected because of their similarities in length and diameter. The lengths and diameters of implants in Groups A, B, and C were 3.0 mm/11.5 mm, 3.0 mm/11 mm, and 3.1 mm/11.5 mm, respectively (Figure 1). In each group, a total of 80 implant placements were completed. Each group was sub-divided into two subgroups (n = 40 per subgroup) depending upon the type of bone (dense or soft (Sawbones, Pacific Research Laboratories Inc., Vashon, WA, USA)) as classical bone simulants for biomechanical testing exhibiting similar properties to bone. Specifically, solid (dense, 40 PCF) foam and cellular (soft, 20 PCF) foam blocks with dimensions 130 mm × 180 mm × 40 mm were used. 

In each subgroup, implant placements were conducted at 800 and 2000 RPM (n = 20 per RPM subgroup) under copious irrigation to avoid risks of overheating. Figure 2 shows a schematic of the distribution of NDIs in each group. All implants were placed under copious irrigation by an experienced, calibrated, and trained periodontist and oral surgeon (GER). The number of the osteotomies was determined based on previous studies from our group [3].

### 2.2. Evaluation of Primary Stability

Immediately after implant placement, PS was assessed using a resonance frequency analysis (RFA) device (Osstell mentor, Integration Diagnostics AB, Gothenburg, Sweden) and recorded as the implant stability quotient (ISQ). A trained and calibrated investigator (D.B.) (kappa value: 0.92) assessed the PS.

### 2.3. Statistical Analysis

Statistical analysis was performed using a computer-based software program (Minitab 16 Minitab Inc., State College, PA, USA). For all groups, the mean values of the ISQ were statistically compared using the one-way analysis of variance with Tukey’s post hoc tests. P-values less than 0.05 were considered statistically significant. 

## 3. Results 

### 3.1. Implant Stability Quotient Values for Implants Placed in Dense Bone at 800 and 2000 PRM

In Groups A, B, and C, the mean ISQ of NDIs inserted at 800 and 2000 RPM were 66.8 ± 3.3, 67.3 ± 2.6, 67.1 ± 2.0, and 65.9 ± 4.2, 66.8 ± 4.9, 63.8 ± 3.1, respectively. In Groups A and B, there was no significant difference in the ISQ for implants inserted at 800 and 2000 RPM. In Group C, the ISQ was significantly higher for NDIs inserted at 800 PRM compared to 2000 RPM (P < 0.05) (Table 1).

### 3.2. Implant Stability Quotient Values for Implants Placed in Soft Bone at 800 and 2000 PRM

In Groups A, B, and C, the mean ISQ of NDIs inserted at 800 and 2000 RPM were 59.9 ± 2.0, 62.4 ± 2.9, 57.9 ± 4.6, and 63.6 ± 1.6, 61.2 ± 3.0, 59.1 ± 2.0, respectively. In Groups B and C, there was no significant difference in the ISQ for implants inserted at 800 and 2000 RPM. In Group A, ISQ values were significantly higher for NDIs inserted at 2000 RPM compared to 800 RPM (P < 0.05) (Table 1).

## 4. Discussion

In the present in vitro experiment, it was hypothesized that there is no statistically significant impact of DS on the PS of NDIs with varying thread designs placed in dense and soft simulated bone. The present results are partially in agreement with the proposed hypothesis, as there was no statistically significant difference in the ISQ values in Groups A and B for NDIs in dense bone, nor in Groups B and C for NDIs inserted in soft bone. These results agree with a previous in vitro study [3] which showed that drilling speeds of 800 and 1500 PRM do not influence PS in dense and soft bone. However, the present experimental study showed two interesting findings. Firstly, PS was significantly higher for NDIs in Group C placed in dense bone at 800 RPM compared to NDIs inserted in the same bone type at 2000 RPM. Secondly, PS was significantly higher for NDIs in Group A placed in soft bone at 2000 RPM compared to that placed at 800 RPM, independent of the recommendations by the manufacturers. 

These differences may be associated with the differences in the tested thread design of the NDIs. Specifically, it seems that symmetrical threads (Group C) improve the implant PS when the used DS is low (800 RPM), and implants with progressive threads have a better PS in soft bone when they are placed using a high DS. Since the used artificial bone has a similar structure and there are consistent stability values due to the low standard deviation for all tested implant designs, using copious irrigation during drilling, there is a practical recommendation for clinicians who use NDIs. According to the bone quality during the pilot drill, clinicians can increase implant stability using a high DS in the case of inconsistent thread geometry and tapered macro-design. This may also be interesting and a first clinical recommendation in orthodontic NDIs or when clinicians have to deal with poor bone qualities. It is recommended that further comparisons between ISQ values and other tests evaluating the mechanical stability of dental implants in order to make a definite conclusion and provide strong clinical recommendations for using (or not) NDIs. Certainly, the clinical condition changes when soft bone quality is modified at the implant interface using bone condensation techniques or bone grafting materials. The appropriate surgical training is fundamental in order to draw conclusions about the impact of the macro-design on the implant PS.

## 5. Conclusions

Within the limits of the present experiment, it is concluded that for NDIs, a lower drilling speed in dense artificial simulated bone and a higher drilling speed in soft artificial simulated bone is associated with high primary stability.

## Figures and Tables

**Figure 1 materials-12-01350-f001:**
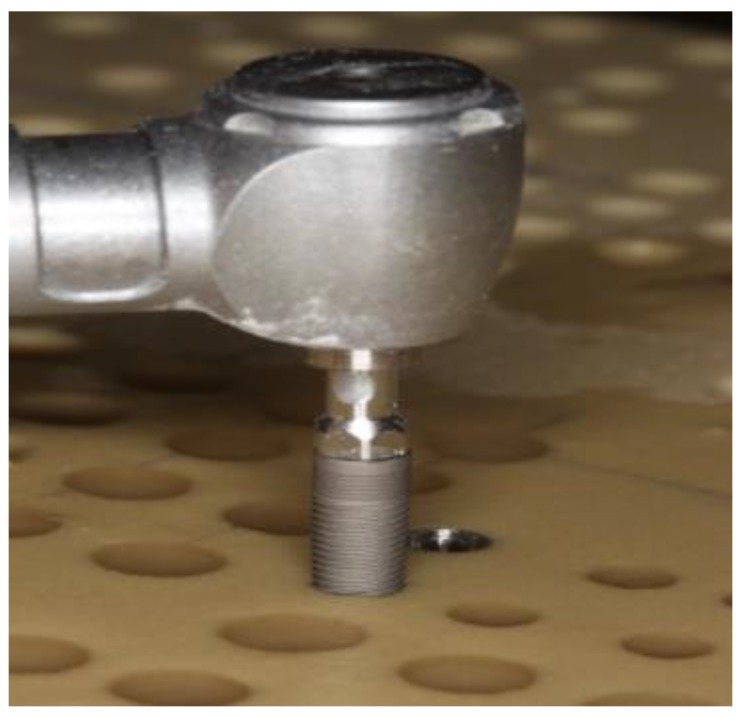
Implant placement in dense bone simulant.

**Figure 2 materials-12-01350-f002:**
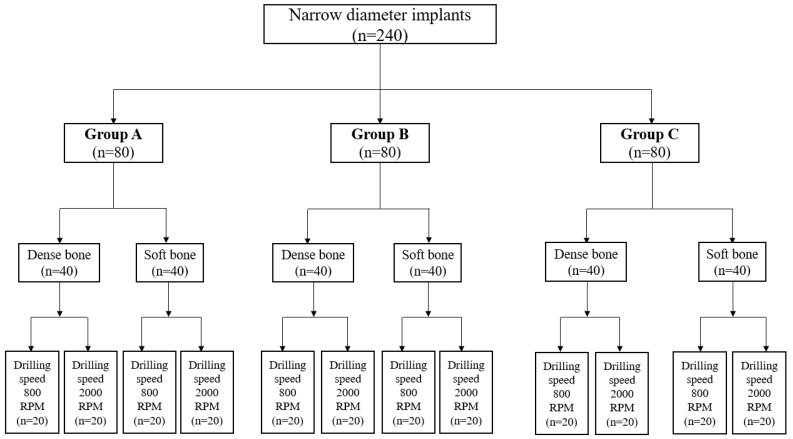
A schematic of the distribution of narrow diameter implants placed in dense and soft artificial bone in Groups A, B, and C.

**Table 1 materials-12-01350-t001:** Mean ± standard deviation of the implant stability quotient for narrow diameter implants placed in dense and soft artificial bone at 800 and 2000 RPM.

Drilling Speed	ISQ Values in Dense Artificial Bone			ISQ Values in Soft Artificial Bone		
	Group A	Group B	Group C	Group A	Group B	Group C
800 RPM	66.8 ± 3.3	67.3 ± 2.6	67.1 ± 2.0 *	59.9 ± 2.0 ^†^	64.4 ± 2.9	57.9 ± 4.6
2000 RPM	65.9 ± 4.2	66.8 ± 4.8	63.8 ± 3.1	63.3 ± 1.6	61.2 ± 3.0	59.1 ± 2.0

ISQ: Implant stability quotient. RPM: Revolutions per minute. * Compared with narrow diameter implants placed in Group C at 2000 RPM (P < 0.05). ^†^ Compared with narrow diameter implants placed in Group A at 2000 RPM (P < 0.05).

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
