# Peer review of "In Vitro Effect of Drilling Speed on the Primary Stability of Narrow Diameter Implants with Varying Thread Designs Placed in Different Qualities of Simulated Bone"

_materials, 2019, doi:10.3390/ma12081350_

Round 1

Reviewer 1 Report

This in-vitro study presents an interesting finding in respect to drilling speed and bone density when placing narrow diameter implants.

The study is well written and the references are adequate.

I have one major issue with the study design. Why were drilling speeds of 800 and 2000 rpm chosen?

The authors mention in the introduction that 'osteotomies performed at higher drilling speeds (1500 rpm) have been associated with thermal changes that may jeopardise the integrity of peri-implant bone tissue.'

One of the implant systems used in the study (Osseospeed EV) recommends the maximum drilling speed of 1500 rpm.

Also, it would be better to have some images to supplement the text.

Author Response

Reviewer 1:

This in-vitro study presents an interesting finding in respect to drilling speed

and bone density when placing narrow diameter implants.

Authors’ response: thank you for the vigilant review.

The study is well written and the references are adequate.

Authors’ response: thank you for the critical evaluation.

I have one major issue with the study design. Why were drilling speeds of 800

and 2000 rpm chosen?

Authors’ response: Thank you for the comment. Drilling speed of 800 rpm was selected as it is the standard of care in implant therapy. Drilling speed of 2,000 rpm was selected as it is the highest speed of the device used to perform the osteotomy.

The authors mention in the introduction that 'osteotomies performed at higher

drilling speeds (1,500 rpm) have been associated with thermal changes that

may jeopardize the integrity of peri-implant bone tissue.'

Authors’ response: We agree that using in clinical situations, drilling speeds of 1,500 rpm and above may jeopardize bone; however, the objective of this study was to assess the stability of narrow diameter implants in artificial bone blocks placed at varying drilling speeds.

© 1996-2019 MDPI (Basel, Switzerland) unless otherwise stated

One of the implant systems used in the study (Osseospeed EV) recommends

the maximum drilling speed of 1,500 rpm.

Authors’ response: The present study had an experimental design in order to assess the influence of drilling speeds on implant stability. By no means do the authors contemplate to present a clinical scenario.

Also, it would be better to have some images to supplement the text.

Authors’ response: We included this time some photo from the experimental setting.

Reviewer 2 Report

Dear Authors, below are my comments about the submitted manuscript

1. Did you use a sample size test to know the amount of specimen you needed for the test or did you base on similar studies? Please clarify.

2. Please explain better and wider the nature of bone you used for the experiment. Was it real bone or bone simulation? And if it was bone simulation, please report its characteristics and not only its provenience.

3. Please clarify the exact thread design of each type of implant used and why you choose specifically those three types and brands.

4. Did you use the same drilling machine for all groups or different machines? Please also report the feature of the machine used.

5. In the Conclusion section, I suggest to be more prudent in speculation: specify that the results are in light of the limitation of the study, do not make suggestion on clinical procedures since it is an in vitro study.

6. Please follow the journal guidelines to format the references style.

After these proper amendments, I encourage re-submission.

Author Response

Reviewer 2:

1. Did you use a sample size test to know the amount of specimen you needed

for the test or did you base on similar studies? Please clarify

Authors’ response: The present study had an experimental design based on previous studies from our group (Almeida et al.)

2. Please explain better and wider the nature of bone you used for the

experiment. Was it real bone or bone simulation? And if it was bone simulation,

please report its characteristics and not only its provenience.

Authors’ response: In the revised manuscript, we have clarified that simulated soft and hard bones were used.

3. Please clarify the exact thread design of each type of implant used and why

you choose specifically those three types and brands.

Authors’ response: These implant designs were selected because of the similarities in lengths and diameters.

996-2019 MDPI (Basel, Switzerland) unless otherwise stated

4. Did you use the same drilling machine for all groups or different machines?

Please also report the feature of the machine used.

Authors’ response: We included this time in the manuscript this information.

5. In the Conclusion section, I suggest to be more prudent in speculation:

specify that the results are in light of the limitation of the study, do not make

suggestion on clinical procedures since it is an in vitro study.

Authors’ response: The conclusion has been revised in the revised manuscript.

6. Please follow the journal guidelines to format the references style.

After these proper amendments, I encourage re-submission.

Authors’ response: The references have been revised as per the Journals’ style.

Reviewer 3 Report

The research article “In vitro effect of drilling speed on the primary 2 stability of narrow diameter implants with varying 3 thread designs placed in different bone qualities” compared three narrow diameter dental implants with different thread design in two different artificial bone hardness using two osteotomy drilling speed.

This a well written paper describing a well design experiment. The results are clinically significant indicating that osteotomy speed should vary in different bone type and with different thread designs.  However,  I would like authors to respond to the following questions and comments:

1-    Refer and clarify the exact quantitative values of the artificial bone density for soft and hard types used in this experiment.  There are a range of densities of artificial bone that may fall within each category.

2-    There are slight differences among the diameter and length of NDIs used.  Although, very small, but could such differences effected the results.

3-    I believe W&H of Austria currently owns all Osstell’s products.  You may want to update “Integration Diagnostics AB, Gothenburg, Sweden”.

4-    Implant companies often used designated osteotomy drills specific to their design.  How was osteotomy sockets prepared.  How would that influenced the results , if different drill designs were used for different implants.

5-    One would expect to find statistical differences between values of the same NDI placed in soft and dense artificial bone.  The reported values in the table are numerically very close. Were there any statistical differences between soft and dense bone of the same NDI design, same speed.  If not , it may be that  the spread of density for soft and dense artificial bone was not wide enough to show statistical differences.

Author Response

Reviewer 3:

This a well written paper describing a well design experiment. The results are

clinically significant indicating that osteotomy speed should vary in different

bone type and with different thread designs. However, I would like authors to

respond to the following questions and comments:

1- Refer and clarify the exact quantitative values of the artificial bone density

for soft and hard types used in this experiment. There are a range of densities

of artificial bone that may fall within each category.

Authors’ response: Information regarding density of the soft and hard bone blocks has been included here.

2- There are slight differences among the diameter and length of NDIs

used. Although, very small, but could such differences effected the results.

Authors’ response: Based upon the reported implant stability values reported in the present experiment, the role of minor variations in implant dimensions seems insignificant.

3- I believe W&H of Austria currently owns all Osstell’s products. You may

want to update “Integration Diagnostics AB, Gothenburg, Sweden”.

Authors’ response: The company details have been updated as recommended.

4- Implant companies often used designated osteotomy drills specific to their

design. How was osteotomy sockets prepared. How would that influenced the

results , if different drill designs were used for different implants.

Authors’ response: We included this time in the manuscript the requested information.

5- One would expect to find statistical differences between values of the

same NDI placed in soft and dense artificial bone. The reported values in the

table are numerically very close. Were there any statistical differences between

soft and dense bone of the same NDI design, same speed. If not, it may be

that the spread of density for soft and dense artificial bone was not wide

enough to show statistical differences.

Authors’ response: Information regarding density of the soft and hard bone blocks has been included here.

Reviewer 4 Report

The  study subject is not relevant for the Journal topics.Too many parameters are considered: the drilling speeed, the implant thread design and the different bone quality. The higher drilling speed examinated (2.000 revolutions per minute) no have a real clinical indications. In the paper there is not an indication about the different parameters that charatcterized the thread design (macro-morphology) of the different dental implants used. No scientific information are reported about the bone simulated material used for the osteotomies. The narrow diameter implants are generally used in the dense bone; to give informations for their use in the soft bone is not very interesting for the clinicians. The paper not present any pictures of the experiment and it is not presented following the instructions fo the Authors. The Introduction and the Discussion are too much synthetics without any considerations about the different surgical and biological aspects of the implant site preparation.

Author Response

Reviewer 4:

The study subject is not relevant for the Journal topics. Too many parameters

are considered: the drilling speed, the implant thread design and the different

bone quality.

Authors’ response: Thank you for the comment. The parameters assessed were implant stability, type of simulated bone and drilling speeds. The authors deem that without these parameters it may be difficult to achieve the objectives of the present study.

The higher drilling speed exanimated (2,000 revolutions per minute) no have a real clinical indication.

Authors’ response: Drilling speed of 2,000 rpm was selected as it is the highest speed of the device used to perform the osteotomy. The present experiment intended to assess the influence of high drilling speeds (such as 2,000 rpm) on the primary stability of narrow diameter implants placed in soft and hard simulated bone. By no means does this experiment suggest increasing drilling speeds up to 2,000 rpm in clinical scenarios.

In the paper there is not an indication about the different parameters that characterized the thread design (macro-morphology) of the different dental implants used. No scientific information is reported about the bone simulated material used for the osteotomies.

Authors’ response: Thank you for the comment. The method of osteotomy, such as the type of surgical motor and bone related parameters such as bone density of the simulated bone used, have been included in the manuscript this time.

The narrow diameter implants are generally used in the dense bone; to give information for their use in the soft bone is not very interesting for the clinicians. The paper not present any pictures of the experiment and it is not presented following the instructions of the Authors.

Authors’ response: Clinical images are now included.

The Introduction and the Discussion are too much synthetics without any considerations about the different surgical and biological aspects of the implant site preparation.

Authors’ response: Information regarding density of the soft and hard bone blocks has been included.

Round 2

Reviewer 1 Report

The required changes have been made. 

Author Response

Thank you

Reviewer 2 Report

Dear Auhors,
good job in making the appropriate modifications.

Author Response

Thank you

Reviewer 4 Report

The introduction is very short. Many considerations can be reported about the drilling speed and its effects in implant site preparation, but they are not evaluated in the discussion. There are not an adequate number of references. 

Author Response

The introduction has been revised and references have been updated

Round 3

Reviewer 4 Report

the discussion and the references were improved. the aim and the research design are more clear now.